# Pathological-Gait Recognition Using Spatiotemporal Graph Convolutional Networks and Attention Model

**DOI:** 10.3390/s22134863

**Published:** 2022-06-27

**Authors:** Jungi Kim, Haneol Seo, Muhammad Tahir Naseem, Chan-Su Lee

**Affiliations:** 1Department of Automotive Lighting Convergence Engineering, Yeungnam University, Gyeongsan 38541, Korea; poui3737@ynu.ac.kr; 2Research Institute of Human Ecology, Yeungnam University, Gyeongsan 38541, Korea; haneol@yu.ac.kr (H.S.); nmtahir@yu.ac.kr (M.T.N.); 3Department of Electronic Engineering, Yeungnam University, Gyeongsan 38541, Korea

**Keywords:** graph convolutional networks (GCN), gait classification, spatiotemporal graph convolutional networks (ST-GCN), multiple-input branches (MIB), global average pooling (GAP), temporal convolutional network (TCN)

## Abstract

Walking is an exercise that uses muscles and joints of the human body and is essential for understanding body condition. Analyzing body movements through gait has been studied and applied in human identification, sports science, and medicine. This study investigated a spatiotemporal graph convolutional network model (ST-GCN), using attention techniques applied to pathological-gait classification from the collected skeletal information. The focus of this study was twofold. The first objective was extracting spatiotemporal features from skeletal information presented by joint connections and applying these features to graph convolutional neural networks. The second objective was developing an attention mechanism for spatiotemporal graph convolutional neural networks, to focus on important joints in the current gait. This model establishes a pathological-gait-classification system for diagnosing sarcopenia. Experiments on three datasets, namely NTU RGB+D, pathological gait of GIST, and multimodal-gait symmetry (MMGS), validate that the proposed model outperforms existing models in gait classification.

## 1. Introduction

Walking is a prevalent behavior that cannot be left out of life, from simple movements to exercise for health. Gait is an essential factor for analyzing body information because it uses the entire body and affects it. Therefore, several studies have been conducted to analyze body-movement information through gait, which have been applied in human identification [1,2,3], sports science [4,5], and medicine [6,7,8].

In medicine, walking is an important factor because it is an exercise that requires the muscles and joints of the entire body. Research is actively conducted on the relationship between walking characteristics, diseases affecting the musculoskeletal system, and physical motor functions such as sarcopenia [9,10]. In computer vision, research is being actively conducted on gait-information collection and pathological-gait classification through sensors or image processing [11,12,13].

Pathological walking is an abnormal walking condition caused by a deformation of the gait or a restriction of the body′s motor function, to reduce pain caused by musculoskeletal or nervous-system abnormalities such as injuries and pain. For example, if labor pain occurs due to an ankle sprain, the leg limps to reduce the pain, and rheumatoid arthritis occurs in knees that cannot bend the leg and walk while swinging it. A pathological gait can appear differently depending on the symptoms, and the symptoms can be diagnosed through gait analysis. We need a spatiotemporal-skeleton model with an applied attention mechanism, to perform pathological-gait classification from the collected skeletal information. To address this, firstly, we need a model that extracts spatiotemporal features from skeletal information represented by joint connections, and, secondly, we need to graph convolutional neural networks using an attention mechanism, to focus on significant parts in the current gait classification. The main inspiration behind applying the pathological-gait classification model is for the diagnosis of sarcopenia.

Sarcopenia is caused by muscle-strength reduction that occurs naturally as humans age. It can cause a decrease in physical-activity ability, difficulty in maintaining daily life functions, and an increase in the risk of accidents such as falls. At a time when population aging is emerging as a social problem, sarcopenia shows a considerable prevalence, and it is considered necessary to pay attention to it. On average, around the globe, it is estimated that 5%–13% of elderly people aged 60–70 years are affected by sarcopenia. The numbers increase to 11%–50% for those aged 80 or above. According to a paper published in the Journal of the American Medical Directors Association in 2020, the prevalence of sarcopenia in Korea was found to be 21.3% for men and 13.8% for women, by applying the 2019 Asian sarcopenia guidelines. Among them, severe sarcopenia was 6.4% in men and 3.2% in women. Since this ratio is so huge, our thrust was to target the problem of sarcopenia, which might assist elderly people with better diagnosis of sarcopenia, using a convenient noninvasive computer-vision technique.

This paper proposes a spatiotemporal feature-analysis model that extracts features for frame-wise-collected human poses from images and classifies pathological gait. Spatiotemporal-feature extraction applies a spatiotemporal-skeleton model with graph convolutional networks (GCN). The skeletal information has joint connections represented as a graph, and features are extracted using a graph convolutional neural network. Computation is performed for additional information: bone expressed by joint connection and velocity (movement) represented by position change, according to the frame. Subsequently, these are used as multiple inputs. The spatiotemporal-attention mechanism focuses on temporal and spatially meaningful joints in gait classification. The proposed model was validated and compared with the state-of-the-art schemes in the literature for NTU RGB+D, pathological gait, and the MMGS datasets, and it provides outstanding accuracy. 

The remainder of this paper is organized as follows. Section 2 describes recent studies related to our work and discusses the limitations of the related work and contributions of the proposed work. Section 3 presents details of the proposed method. Extensive experiments on three datasets are reported in Section 4, with visualization and analysis of pathological-gait data, and Section 5 concludes the study.

## 2. Related Work

### 2.1. Gait Recognition

Several sensor- and vision-based methods for gathering walking information for gait recognition are available [14]. Sensor-based methods are not significantly affected by the surrounding environment. Their drawback is that only limited information can be collected, while vision-based methods are relatively rich in information but are affected by surroundings such as backgrounds and occlusion [15].

According to the body expressions for gait recognition, vision-based methods are divided into silhouette-based [16,17,18] and skeleton-based methods [19,20,21,22,23]. Silhouette-based methods are prominent in gait recognition, perform human identification, can be easily calculated through image binarization, and provide an outline of the target, thereby identifying the person or a carrier state. However, it is challenging to change the viewpoint and, therefore, they are limited by the obtainable body information. Skeleton-based gait recognition methods are robust to viewpoint changes and can, thus, obtain more body information. However, these methods are affected by pose-estimation performance and are, therefore, computationally expensive.

Researchers in [17] applied the horizontal-bins method to divide one silhouette into several parts to extract features and learn walking features according to the region. The gait lateral network was trained in a previous study [18], by applying pooling to the entire silhouette bundle of each frame′s gait sequences and silhouettes. Another model, a mixture of a convolutional neural network (CNN) and a long short-term memory (LSTM) network, was trained by extracting spatiotemporal features from the joint-position information, as discussed in [19].

Jun et al. [23] performed pathological-gait classification by inputting skeletal information into the gated-recurrent-unit (GRU) network, identifying functional body parts by categorizing them, and comparing their performance when trained only with certain aspects.

### 2.2. Graph Convolutional Network

Graph convolutional neural network is a type of neural network that learns data represented by the graphs as input [24,25]. The graph consists of edges connecting nodes and neighboring nodes. A node refers to data, and an edge refers to a relationship between data. It is a method of expressing a graph as an adjacent matrix and a feature matrix, multiplying it by a weight matrix to update it. The adjacent matrix A is defined by the edge E of the graph, as shown in Equation (1). A normalized adjacency matrix is obtained by adding self-regression to include its features and going through normalization, so that each node is unaffected by the number of neighboring nodes. Equation (4) represents the state of each node, calculated using an adjacent matrix, as shown in Equation (6).
(1)$$A_{i,j}=\left\{\begin{array}{c} 1\;\left(v_{i},v_{j}\right)\in E\\ 0\;\left(v_{i},v_{j}\right)\notin E \end{array}\right.,$$
(2)$$\tilde{A}=A+I,$$
(3)$$A_{norm}=D^{-\frac{1}{2}}\tilde{A}D^{-\frac{1}{2}},$$
(4)$$H_{i}^{\left(l+1\right)}=\sigma \left(H_{0}^{\left(l\right)}W_{0}^{\left(l\right)}+H_{1}^{\left(l\right)}W_{1}^{\left(l\right)}+H_{2}^{\left(l\right)}W_{2}^{\left(l\right)}+\ldots +b^{\left(l\right)}\right),$$
(5)$$H_{i}^{\left(l+1\right)}=\sigma \left(\sum_{j\in N\left(i\right)}H_{j}^{\left(l\right)}W^{\left(l\right)}+b^{\left(l\right)}\right),$$
(6)$$H_{i}^{\left(l+1\right)}=\sigma \left(AH^{\left(l\right)}W^{\left(l\right)}+b^{\left(l\right)}\right).$$

Body-skeleton information is suitable for expressing as a graph because joints can be represented as nodes and edges. Thus, it can be used to analyze body movements, including action recognition [26,27,28,29]. Researchers of [26] used a spatial-graph-convolution operation and temporal-convolution operation to allow spatiotemporal features to be considered and weight connections to be made according to the joint position. However, the method could not consider the entire body in action recognition because it only considered the relationship between joints within the designated connection range. Researchers of [27] added an A-link to the structure of the method, as described in [26]. Two types of methods were applied to connect the nodes: S-link, which express the relationships between adjacent joints, and A-link, which expresses the relationships between distant joints. A limitation of the A-link-extraction process is that the values were combined into one, according to the time axis, so changes over time were not sufficiently considered. Another study [28] defined a part-specific graph-convolutional-network model, by specifying a bundle of each joint and defining a part-based graph.

### 2.3. Attention Mechanism

The attention mechanism focuses on meaningful parts without referring to all inputs simultaneously. The attention mechanism shows significant advances in computer vision, starting with a recurrent attention model (RAM)-based [30] recurrent neural network (RNN) and reinforcement learning, which is used in image classification [31,32], object recognition [33,34], face recognition [35,36], pose estimation [37], action recognition [38,39], and medical-image processing [40,41].

Squeeze-and-excitation networks (SENet) [31] are attention models that can be applied to existing models to improve their performance. As the name “Squeeze-and-Excitation Networks” suggests, the squeeze phase and fully connected (FC) layers determine the importance of each channel. It has a straightforward structure; therefore, it does not increase the model complexity, and the model performance improvement is larger than the increase in parameters.

The bottleneck-attention module (BAM) [42] is located in the bottleneck part of the existing model, such as the bottleneck-attachment module. Channel attention with GAP and spatial attention with dilated convolution are combined in parallel. The convolutional block-attention module (CBAM) [43] is a follow-up study of the one that proposed BAM [42], which uses AvgPooling. CBAM uses MaxPooling and AvgPooling in combination, and channel pooling is connected in series but not in parallel. Coordinate attention for efficient mobile-network design, as discussed in [44], performs channel attention while maintaining spatial information by pooling in the H and W directions, to preserve the spatial information in a 2D-feature map.

### 2.4. Limitations of Related Work and Contributions

Table 1 summarizes the problems associated with the existing approaches. The previous methods have at least one of the following weaknesses.
Although the models discussed in [12,23] provided outstanding accuracy, they were tested on less diverse datasets.Although the model in [24] used diverse datasets, it had low accuracy.The work in [45] experimented with a small number of classes.Dependence on a fixed set of handcrafted features requires deep knowledge of the image characteristics [27]. They rely on texture analysis, where a limited set of local descriptors computed from an image is fed into classifiers such as random forests. Despite the excellent accuracy in some studies, these techniques are limited in terms of generalization and the transfer capabilities are limited in terms of inter-dataset variability.Inefficient algorithms result in higher computational costs and time [46,47].Although the models discussed in [29,48] provide outstanding accuracy, they cannot fuse RGB modalities and different skeleton sequences with object appearance.

**Table 1 sensors-22-04863-t001:** Comparison and weaknesses of related work.

Publications	Method	Dataset	Accuracy	Weakness
Khokhlova et al. [12]	Using single LSTM Ensemble LSTM	MMGS dataset	94 91	Lack of diverse datasets
Jun et al. [23]	Using GRU	Newly created dataset	93.7	Lack of diverse datasets
Yan et al. [24]	Using spatiotemporal GCN	Kinematics + NTU-RGBD	88.3	Less accuracy
Liao et al. [27]	Using CNN	CASIA B + CASIA E	-	Uses few handcrafted features
Shi et al. [29]	Using GCN	NTU-RGB + Kinematics skeleton	90	Unable to fuse RGB modality
Lie et al. [49]	Using pose-refinement GCN	Kinematics + NTURGB-D	91.7	Less accuracy
Ding et al. [45]	Using Semantics guided GCN	NTU + Kinetics	94.2	Use a smaller number of classes
Song et al. [46]	Using multi-stream GCN	NTU RGB-D 60 + NTU RGB-D 120	82.7	Network complexities
Shi et al. [47]	Using two-stream adaptive GCN	NTU RGB D + Kinetics	95.1	Network complexities
Si et al. [48]	Using attention-enhanced GC LSTM	NTU RGB D + North-Western UCLA	93.3	Unable to fuse skeleton sequence with object appearance

Some of the previous models discussed in the aforementioned papers give outstanding accuracy. Still, they were tested on less-diverse datasets, while some used large, various datasets and gave low accuracy. Contrary to previous works, the proposed approach does not rely on a semi-automatic process for feature selection, but computes all features automatically. This paper presents the following major contributions.
A spatiotemporal graph convolutional network (ST-GCN) using attention models from skeletal information is proposed to extract spatiotemporal features presented by joint connections and applied to pathological-gait classification.A fused model, receiving inputs from three separate spatiotemporal-feature sequences (joint, velocity, and bone) obtained from raw skeletal data, shows an improvement in the performance of the pathological-gait classification over other skeletal features.Diverse datasets, such as NTU RGB+D, pathological gait, and MMGS data, are used to evaluate the proposed model and show better performance than other deep-learning-based approaches.For the NTU RGB+D, GIST and MMGS datasets, the proposed multiple-input model with an attention model gives better performance than other existing schemes.

## 3. Datasets and Methods

This section describes the proposed ST-GCN with an attention model for gait classification. First, the three datasets used in this study are described. Then, the preprocessing of the datasets is presented. Finally, the proposed model is described in detail: pathological-gait recognition using spatiotemporal graph convolution networks and an attention model. 

### 3.1. Datasets

NTU RGB+D [44], GIST: pathological gait [20], and MMGS [12] datasets are used in the experiments. All datasets were collected using a Microsoft Kinect V2 camera. The NTU RGB+D dataset contains RGB and infrared images, depth maps, and three-dimensional skeletal data. Pathological gait and MMGS consist only of three-dimensional skeletal data.

#### 3.1.1. NTU RGB+D Dataset

Figure 1 shows a few sample images from the NTU RGB+D dataset. The NTU RGB+D dataset captures 60 actions observed daily for action recognition. Forty subjects performed 60 types of actions that were captured from the front, side, and 45° diagonal directions using three cameras. The total number of image samples was 56,800, and the training and test sets were divided by the number of subjects. 

#### 3.1.2. GIST: Pathological-Gait Dataset

The pathological-gait dataset had one normal gait and five abnormal gaits for pathological-gait classification, as shown in Figure 2. Ten subjects performed six types of walks, and six cameras collected the three-dimensional skeletal information. The total number of samples was 7200, and, in this experiment, the training and test sets were divided using the leave-one-subject-out cross-validation method.

#### 3.1.3. MMGS Dataset

The MMGS dataset contained one normal gait and two abnormal gaits for pathological-gait classification. Twenty-seven subjects performed three types of walking, and six cameras collected the three-dimensional skeletal information. The total number of samples was 475, and the training and test sets were divided by subject number.

### 3.2. Preprocessing

Data preprocessing is essential for skeleton-based action recognition. In this work, the input features after various preprocessing steps were mainly divided into three classes: (1) joint positions, (2) motion velocities, and (3) bone features, which had three-dimensional coordinates obtained from the Kinect camera and relative coordinates calculated from the center joint. The motion velocity feature was calculated from the change in joint position per frame. Bone features use vectors, and their angles are calculated by considering joint connections.

The joint features use joint coordinates and relative coordinates, and the relative coordinates are calculated as in Equation (7). Velocity features use one or two frames to change the position by a specified amount. The bone feature calculates the bone vector through the position difference with adjacent joints, and the vector angle using the inverse trigonometric function.
(7)$$r_{i}=x_{i}-x_{c}$$
(8)$$v_{t1}=x_{t+1}-x_{t},$$
(9)$$v_{t2}=x_{t+2}-x_{t},$$
(10)$$b_{i}=x_{i}-x_{adj},$$
(11)$$a_{i}=\arccos\left(\frac{b_{i}}{\left\|b_{i}\right\|}\right).$$

### 3.3. Pathological-Gait Recognition Using Spatiotemporal Graph Convolution Networks and Attention Model

Figure 3 shows the entire pipeline of the proposed model, where the three input sequences (joint, velocity, and bone) were initially extracted from the original skeleton sequence. Each feature was input into the ST-GCN layer separately. The ST-GCN layers have an ST-GCN block and an attention block (as shown in Figure 4), extracting spatiotemporal features from the ST-GCN block and applying the attention mechanism. The outputs of each branch were fused and input into the mainstream. Finally, it was entered into a classifier composed of a global average pool (GAP) and fully connected (FC) layers, to classify gait.

After data preprocessing, we obtained three types of input data: joint, velocity, and bone. Current high-performance complex models typically use multiple-input architectures to handle these inputs. For example, Shi et al. [47] used joint and bone data as inputs for feeding to two GCN branches with similar model structures separately, and eventually chose the fusion results of two streams as the final decision. This effectively augmented the input data and enhanced model performance. However, a multistream network often entails high computational costs and difficulties in parameter tuning on large-scale datasets. We used the multiple-input branch (MIB) architecture that fuses the three input branches and then applies one mainstream to extract discriminative features. This architecture retains the rich input features and significantly suppresses the model complexity with fewer parameters; thus, it is easier to train. An example of the proposed ST-GCN with MIB model is shown in Figure 4. 

The input branches were formed by orderly stacking a BatchNorm layer for fast convergence, an initial block implemented by the ST-GCN layer for data-to-feature transformation, and three GCN blocks with attention for informative-feature extraction. After the input branches, a concatenation operation was employed to fuse the feature maps of the three branches and then send them to the mainstream that was constructed using two GCN blocks. Finally, the output feature map of the mainstream was globally averaged to a feature vector, and an FC layer determined the final action class.

Figure 5 (left) shows the basic components of the ST-GCN implemented by orderly stacking a GCN layer and several temporal convolutional (TC) layers. The depth of each GCN block was the number of TC layers stacked in the block. In addition, for each layer, a residual link made the model optimization easier than the original unreferenced feature projection. The first TC layer had a stride of 2 for each block in the mainstream, which compressed the features and reduced the convolutional costs. 

This study used the ST-GCN model as a spatiotemporal-skeleton model. Feature maps were created with features extracted from multiple inputs, and adjacency matrices were defined according to predefined skeletal models [51]. We considered spatial features by applying graph convolutional neural networks and temporal features, by applying convolutional neural networks on a time axis, as shown in Figure 5 (right).

An overview of the proposed ST-GCN module is shown in Figure 6, from which the input features were first averaged at the frame and joint levels. These pooled feature vectors were then concatenated and fed through an FC layer to compact the information. Next, two independent FC layers obtained two attention scores for the frame and joint dimensions. Finally, the scores of the frames and joints were multiplied by the channel-wise outer product, and the results were the attention scores for the whole action sequence. 

### 3.4. Ablation Study

We introduced the bottleneck structure into the ST-GCN model, as shown in Figure 4, for reducing the model size. The proposed model contains three input branches, joints, velocity, and bones, which is shown in Figure 3. Table 2 (in the Experimental Results section) presents the ablation studies of the input data. It clearly indicates that the model with only one input branch is significantly worse than the others. This implies that each input branch is necessary to the model, and our model takes a huge benefit from its multiple-data-inputs structure. 

## 4. Experimental Results

This section presents the evaluation of the proposed ST-GCN on three large-scale datasets: NTU RGB+D, GIST: pathological gait, and MMGS. Ablation studies were performed to validate the contribution of each component to our model. Finally, an analysis of the results is reported to prove the effectiveness of the proposed method.

### 4.1. Multiple Data Inputs

The proposed model contains three input branches, which are shown in Figure 3. To compare the performance according to the input combinations, we show the performance of spatiotemporal-skeletal models for multiple inputs on the NTU RGB+D dataset in Table 2. We determine the best combination with the highest performance for the three inputs: joint, velocity, and bone. The performance was improved using multiple features rather than a single feature, and the multiple-input model using all three inputs showed the highest performance, at 87.7%. Since our proposed model takes a huge benefit from multiple data inputs, three-input models are used to evaluate the performance of the proposed model.

### 4.2. NTU RGB+D Dataset

A performance evaluation was conducted on the NTU RGB+D dataset in comparison with the existing action-recognition models, as shown in Table 3. ST-GCN [24] is currently the most popular backbone model for skeleton-based action recognition, exhibiting an accuracy of 81.5%. To check the validity of multiple input and attention mechanisms, the accuracy of multiple-input ST-GCN was 87.7%, showing an improvement of approximately 6.2% from ST-GCN. The accuracy of multiple-input ST-GCN with the attention mechanism was 89.6%, which further improved the performance of our proposed model. Finally, after observing the results in Table 3, we can conclude that the proposed model performed well, especially for the attention mechanism.

### 4.3. GIST: Pathological-Gait Dataset

Table 4 shows a performance comparison of the proposed model with the existing schemes for the pathological-gait dataset. Jun at el. designed a GRU-based model [23], which showed 90.1% performance when the entire skeleton was used and 93.7% when only the joints of the legs were used as input. The accuracy of ST-GCN was 94.5%, showing an improvement of approximately 4.4% over the existing GRU network. The accuracy of multiple-input ST-GCN was 98.30% and that of multiple-input ST-GCN with the attention mechanism was 98.34%. We can find performance improvement through multiple-input methods and an attention mechanism. In Table 4, we can conclude that the proposed model performed well with or without an attention mechanism.

### 4.4. MMGS Dataset

Table 5 shows a performance comparison of the proposed model with existing schemes for the MMGS dataset. Khokhlova et al. [12] collected a new multimodal gait symmetry (MMGS) dataset that contains skeleton data, including skeleton-joint orientations. They adopted two LSTM models: a single LSTM and an ensemble LSTM. An ensemble-LSTM model was proposed to decrease the variance of each model and its dependency on the test partitioning. The experiments were performed using the collected MMGS database. The accuracy of the multiple-input ST-GCN with the attention mechanism was improved by 1.5% compared to the single LSTM model and by 4.5% compared to the ensemble LSTM model. Finally, after observing the results in Table 5, we can conclude that the proposed model performed well with the attention mechanism.

### 4.5. Skeleton Data Visualization and Gait-Characteristics Analysis

This subsection explains why the ST-GCN method can achieve superior accuracy but with fewer model parameters than the traditional GCN models. Separable convolution was initially designed as the core layers were built, aiming to deploy deep-learning models on computationally limited platforms such as robotics, self-driving cars, and augmented reality. As its name implies, separable convolution factorizes a standard convolution into depth- and point-wise convolutions. Specifically, for depth-wise convolution, a convolutional filter is only applied to one corresponding channel, whereas point-wise convolution uses a 1 × 1 convolution layer to combine the output of depth-wise convolution and adjust the number of output channels, as shown in Figure 7. 

Skeleton weights were visualized using class-activation maps (CAM) [53] to analyze the important joints in gait. If the feature map has *c* channels, then each channel has one representative value through global average pooling, and *c* weights are obtained through the fully connected layer. This weight is applied as a weighted sum to the feature map; it determines which part the model sees and predicts for the corresponding class. If this weight is used as a weighted sum to the feature map, we can find an important part for predicting the class. Jun et al. found a useful body part for pathological-gait classification by comparing the results of training only certain parts (O-series) or excluding some parts (E-series) as a baseline, when all joints were entered in the result analysis of the GRU-based model [23]. However, this study analyzed the skeletal weights using the CAM technique without distinguishing the input parts, confirming the important joints according to class.

#### 4.5.1. Normal Gait

In a normal gait, the spine and pelvis appear to be important joints. The pelvis and spine are activated in the posture of stretching the feet while walking, rather than in a neutral position where both feet are parallel.

#### 4.5.2. Antalgic Gait

Antalgic gait is a type of gait caused by injury to the leg part or pain caused by a particular disease. If weight is put on the symptomatic leg, pain is felt, so the time to step on it is shortened, and the sufferer limps. When the injured leg is placed on the ground, the characteristics of painstaking walking appear, and the legs and pelvis are activated.

#### 4.5.3. Lurch Gait

Lurch gait is a walk caused by pain due to abnormalities in the hip areas, such as weakening or paralysis of the gluteus maximus. When the sufferer steps on the leg, their upper body is laid back, and the leg that is stepped on to balance extends further and stumbles. When the leg on the symptomatic side is extended forward, the leg and pelvic areas are activated.

#### 4.5.4. Steppage Gait

Steppage gait is a walk in which the tip of the foot does not want to step on the ground, and is caused by muscle and motor nerve abnormalities in the front shin. Due to problems associated with ankle bending, the legs are raised high, and the legs, pelvis, and spine are activated when the symptomatic legs are raised high.

#### 4.5.5. Stiff-Legged Gait

Stiff-legged gait, also known as stiff-knee gait, is caused by joint abnormalities in the knee area. The knees cannot be bent and straightened, and they are always stretched out; therefore, the legs are swung in a semicircular shape while walking. The area is activated when swinging a diseased leg.

#### 4.5.6. Trendelenburg Gait

Trendelenburg gait is caused by the weakening or paralysis of the blunt middle force, and the torso tilts in the direction of symptoms when walking. As the pelvis is imbalanced, the upper body is tilted to balance, and it is clear that both the pelvis and shoulders are activated.

CAM visualizes skeletal weights and identifies important joints for pathological-gait classification. In pathological gait, activation can be confirmed in parts showing the characteristic movements of walking due to symptoms.

In pathological-gait classification, joints that are important for judgment are not uniform and appear differently depending on the symptoms. For the pathological-gait-classification problem, it is suitable to use the entire skeleton as an input rather than directly specifying the input, as shown in Figure 3. However, it focuses on the important joints for determining the current symptoms by applying an attention mechanism.

## 5. Conclusions and Future Work

This study proposed a graph-convolutional-neural-network model with multiple inputs and attention techniques for pathological-gait classification for the diagnosis of sarcopenia. The proposed model is suitable for pathological-gait classification, by applying multiple input and attention mechanisms to GCN-based spatiotemporal-skeleton models. The ST-GCN was applied as a skeleton model that could use all spatiotemporal elements, and the spatiotemporal features were extracted considering the joint connections. To validate multiple-input methods and find the best combination of inputs, the accuracy was compared according to the combination of inputs, and all three inputs showed the best performance. The attention mechanism was also configured to apply spatiotemporal attention and showed additional performance improvements when applied to multiple-input models. Finally, through bone-weight analysis, important joints that depend on the symptoms were identified in the pathological-gait classification.

The model was also compared with the state-of-the-art approaches for the NTU RGB+D, GIST and MMGS datasets. For the NTU RGB+D dataset, the model gives the accuracy of 87.8% and 89.6% for multiple input and multiple input with attention, respectively, and those results are higher than the accuracies of other models. Again, for the GIST dataset, the model gives the accuracy of 98.30% for multiple input and 98.34% for multiple input with attention, both of which are also higher than the accuracies of other models. Similarly, for the MMGS dataset, the model gives the accuracy of 95.5% for multiple input with attention, which is again higher than the accuracies of other models. In conclusion, since, for all three datasets, the accuracies of the proposed model are high, our proposed model outperformed the other models.

Experiments in the future will deal with the determination of sarcopenia, based on different analysis of gaits using our datasets or other datasets, in a plan specialized for sarcopenia. Moreover, in the future, we can also make a real-time system for quickly and more efficiently diagnosing sarcopenia. We will also try to improve overall performance by improving the spatiotemporal-attention mechanism. Since the lightening of the model is also an important issue, as it uses multiple inputs, it is planned to apply an appropriate lightening method to configure a light model, while maintaining performance.

## Figures and Tables

**Figure 1 sensors-22-04863-f001:**
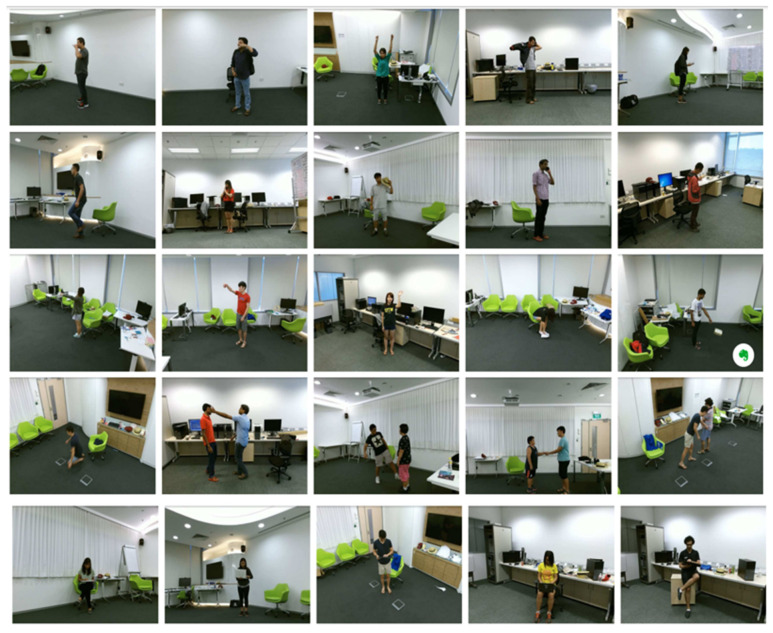
Sample images from NTU RGB+D dataset [50].

**Figure 2 sensors-22-04863-f002:**
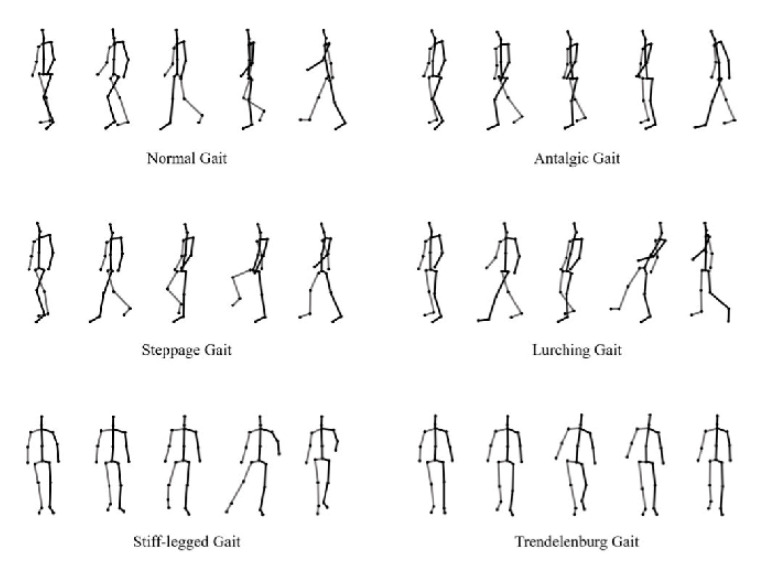
Skeleton data of normal and pathological gaits.

**Figure 3 sensors-22-04863-f003:**
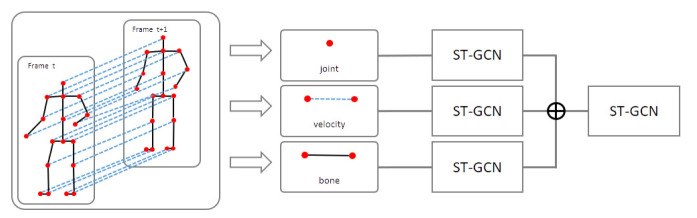
Overall pipeline of our proposed approach.

**Figure 4 sensors-22-04863-f004:**
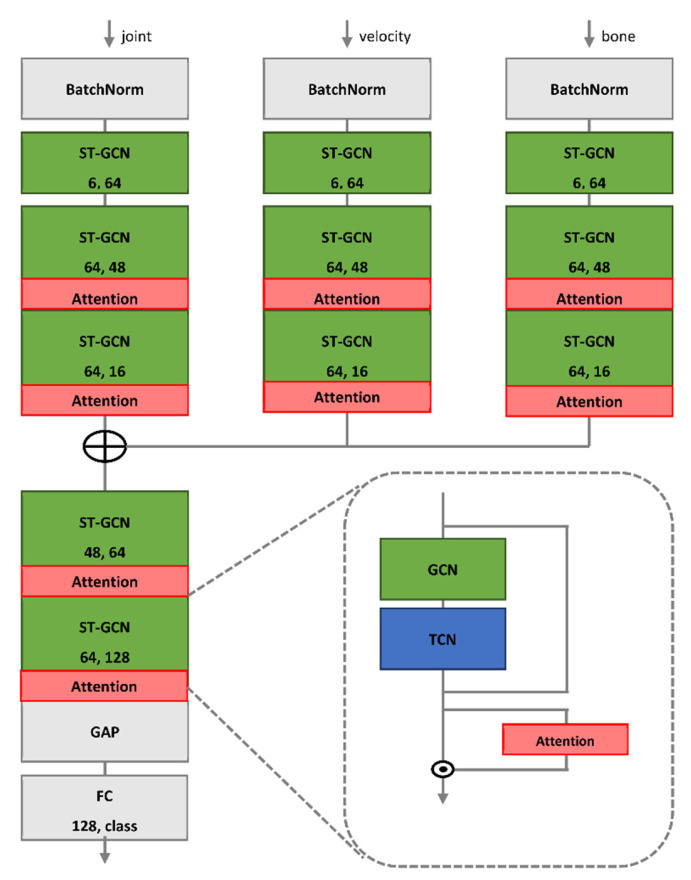
Overview of proposed model. Two numbers in each block denote input and output channels, and ⊕ represents concatenation.

**Figure 5 sensors-22-04863-f005:**
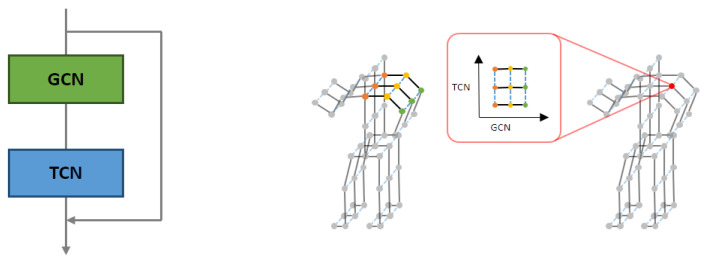
Depth of proposed ST-GCN block: details of ST-GCN (**left**) and spatial features of GCN and TCN (**right**).

**Figure 6 sensors-22-04863-f006:**
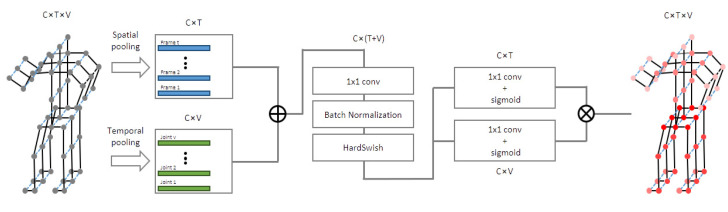
Overview of the proposed ST-GCN module, where C, T, and V denote the numbers of input channels, frames, and joints, respectively.

**Figure 7 sensors-22-04863-f007:**
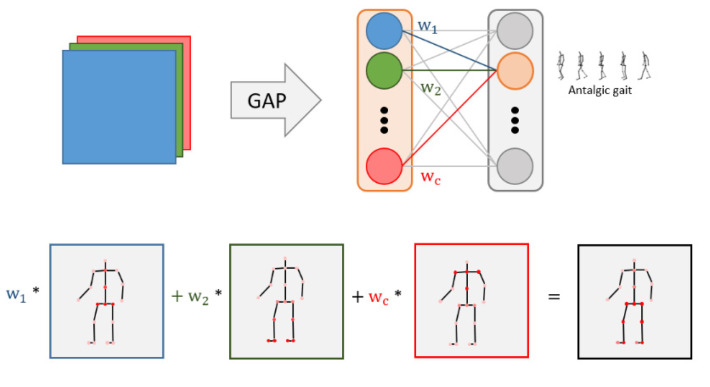
Separable convolution for skeleton-based action recognition.

**Table 2 sensors-22-04863-t002:** Performance for multiple data inputs for joints, velocity, and bones.

Input Type	Accuracy (%)
Joints	82.9
Velocity	82.4
Bones	84.0
Joints + Velocity	86.8
Joints + Bones	85.4
Velocity + Bones	87.3
Joints + Velocity + Bones	**87.7**

**Table 3 sensors-22-04863-t003:** Comparison of performance of the proposed method with the schemes in the literature for NTU RGB+D dataset.

Model	Accuracy (%)
ST-GCN [24]	81.5
PR-GCN [49]	85.2
Sem-GCN [45]	86.2
AS-GCN [27]	86.8
RA-GCN [46]	87.3
PB-GCN [28]	87.5
2s-AGCN [47]	88.5
AGC-LSTM [48]	89.2
Multiple-input ST-GCN	**87.7**
Multiple-input ST-GCN (+Attention)	**89.6**

**Table 4 sensors-22-04863-t004:** Comparison of performance of proposed method with the schemes in the literature for pathological-gait dataset.

Model	Accuracy (%)
GRU (full-skeleton) [23]	90.1
GRU (only legs) [23]	93.7
ST-GCN	94.5
Multiple-input ST-GCN	**98.30**
Multiple-input ST-GCN (+Attention)	**98.34**

**Table 5 sensors-22-04863-t005:** Comparison of performance of the proposed method with the schemes in the literature for MMGS dataset.

Model	Accuracy (%)
Single-LSTM model [12]	94
Ensemble-LSTM model [12]	91
AGS-GCN [52] Multiple-input ST-GCN (+Attention)	**92.3** **95.5**

## Data Availability

Not applicable.

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
