# Peer review of "Pathological-Gait Recognition Using Spatiotemporal Graph Convolutional Networks and Attention Model"

_sensors, 2022, doi:10.3390/s22134863_

Round 1

Reviewer 1 Report

General Comment:

The article is fluent, easily readable, with a large survey on existing methods for comparison.

However, ss a general comment, I find figures 8 to 13 not strictly necessary as they do not provide any information, they are redundant. I bet who would be able of recognizing at first glance which type of gait each individual figure belongs to. The concept is clear, I suggest removing figures 8, 9, 10, 11, 12 and 13 as the individual views are already introduced in the previous figures 2, 3, 5 and 6. Just leave the descriptions of the different types of gait. In addition, the main advantage of the proposed method seems to be the temporal analysis of the inputs that cannot be visible with static figures.
In any case, please comment.

Some tables have some numbers hard to be compared. The accuracy column in Table 1 shows some numbers with 2 digits (94), others use 4 digits (93.67). How to compare these accuracies and why the author use different representation? Same comments for Tables 3 and 5. The authors seem to give emphasis to their methods that are represented in bold and using 4 digits instead of 2, for the Accuracy %. In Table 5, for example, the Single LSTM model  is associated with an accuracy of 94 % while the Multi input ST-GCN with 94.45 % that, after rounding, is still 94. Please, converge on using the same number of digits for all the representations.

Author Response

I would like to say thanks to consider our article and giving beneficial feedback to improve the article in terms of content writing, diagrams and references etc.

According to the suggestions, the article has been reviewed by the authors and tried our level best to meet the entire requirements. Here, there are amendments to the current write-up.

Point 1: However, as a general comment, I find figures 8 to 13 not strictly necessary as they do not provide any information, they are redundant. I bet who would be able of recognizing at first glance which type of gait each individual figure belongs to. The concept is clear, I suggest removing figures 8, 9, 10, 11, 12 and 13 as the individual views are already introduced in the previous figures 2, 3, 5 and 6. Just leave the descriptions of the different types of gait. In addition, the main advantage of the proposed method seems to be the temporal analysis of the inputs that cannot be visible with static figures.
In any case, please comment.
Response 1: According to your valuable suggestions, we have removed the figures from 8 to 13 while retaining the description in the revised manuscript.

Point 2: Some tables have some numbers hard to be compared. The accuracy column in Table 1 shows some numbers with 2 digits (94), others use 4 digits (93.67). How to compare these accuracies and why the author use different representation? Same comments for Tables 3 and 5. The authors seem to give emphasis to their methods that are represented in bold and using 4 digits instead of 2, for the Accuracy %. In Table 5, for example, the Single LSTM model is associated with an accuracy of 94 % while the Multi input ST-GCN with 94.45 % that, after rounding, is still 94. Please, converge on using the same number of digits for all the representations.

Response 2: Thanks for your suggestion. We have changed the representation in Table 1, Table 3, and Table 5 to 3 digits in the revised manuscript whenever it is possible. In addition, we have also used a similar representation for the other tables like Table 2 and Table 4 in the revised manuscript. For our experiment results with and without the attention model, we used 4 digits to show the difference in the performance.  

Reviewer 2 Report

The manuscript discusses an improved model for gait classification. The manuscript is well written with a smooth flow for the most part. However, the authors are suggested to improve the manuscript by considering following points:

1)      There are some instances where the writing could be improved. For example:

a.       In the beginning of para 2 in Introduction, there is no use for the word ‘also’.

b.       Page 5, line 196: Perhaps an “are” is missing.

2)      The figures are a bit pixelated. Could the authors please put in high-res figures?

3)      I suggest changing section 2 `Related Work´ to Experimental Section

4)      Sentence ´Their drawback is that only limited information can be collected, while vision-based methods are relatively rich in information but are affected by surroundings such as backgrounds and occlusion ´ needs suitable reference, I suggest to cite a seminal work on the vision based info on motion analysis with topic DOI: 10.1055/s-0037-1604202 to make reference up to date.

5)      Page 2, line 48: Which model are the authors talking about? Is it the one which is “needed” as stated in the previous sentence? If so, then specifying that the model was indeed developed could be beneficial before specifying what the model does.

6)      How prevalent is sarcopenia? Is there a specific reason why this particular condition is targeted?

7)      Why were the training and test sets in GIST not divided by subject number like the other two? Wouldn’t it be better to have the same division method for all three?

8)      Does Table 2 present results of the model before describing the model itself? Wouldn’t it be better placed in the Results section?

9)      When discussing gaits, what do the authors mean by being activated? For example, what does “the pelvis and shoulders are activated” mean?

10)   The conclusion is a bit confusing. There are many tables that talk about the accuracy of the model developed in the paper but that is not mentioned in the conclusion. This makes the reader doubt as to what those accuracies were.

11)   Cite a recent report https://doi.org/10.1002/adhm.201901862 on the topic with sentence ´Feature maps 302 were created with features extracted from multiple inputs, and adjacency matrices were defined according to predefined skeletal models ´

12)   The authors say that “The main inspiration behind applying the pathological gait classification model is for the diagnosis of sarcopenia” but in the end sarcopenia is never identified. If it is a future work, then the authors are suggested to not mention it in future work and not make it the focus or the main motivation of the paper.

Author Response

I would like to say thanks to consider our article and giving beneficial feedback to improve the article in terms of content writing, diagrams and references etc.

According to the suggestions, the article has been reviewed by the authors and tried our level best to meet the entire requirements. Here, there are amendments to the current write-up.

The manuscript discusses an improved model for gait classification. The manuscript is well written with a smooth flow for the most part. However, the authors are suggested to improve the manuscript by considering following points:

1)      There are some instances where the writing could be improved. For example:

  1. In the beginning of para 2 in Introduction, there is no use for the word ‘also’.

Response 1a: Thanks for your valuable suggestion. The word ‘also’ is removed from the beginning of para 2 in the Introduction section.

  1. Page 5, line 196: Perhaps an “are” is missing.

Response 1b: Thanks for your valuable suggestion. The word ‘are’ is added on page 5, line 196, in the revised manuscript.  

2)      The figures are a bit pixelated. Could the authors please put in high-res figures?

Response 2: According to your suggestion, Figure 1 in the revised manuscript is changed to high resolution from the original paper, and Figure 8-11 were removed based on the suggestion of another reviwer. Other figures seem fine in our opinion.

3)      I suggest changing section 2 `Related Work´ to Experimental Section

Response 3: Thank you for your suggestion. We discussed the suggestion with other authors and we decided to keep the related work as it is because, in Section 2, we focused on other works related to our study, and in the Experimental results section, we focused on the experiment results of our proposed model and its comparison to other works.

4)      Sentence ´Their drawback is that only limited information can be collected, while vision-based methods are relatively rich in information but are affected by surroundings such as backgrounds and occlusion ´ needs suitable reference, I suggest to cite a seminal work on the vision based info on motion analysis with topic DOI: 10.1055/s-0037-1604202 to make reference up to date.

Response 4: According to your suggestion, the said reference is added in the revised manuscript.

5)      Page 2, line 48: Which model are the authors talking about? Is it the one which is “needed” as stated in the previous sentence? If so, then specifying that the model was indeed developed could be beneficial before specifying what the model does.

Response 5: Thanks for your suggestion. At page 2, line 48, we are basically talking about the model we need to address the problem, a proposed system. Moreover, the statement is made clear like ‘To address this,…’ which actually relates with the previous sentence ‘We need a ……’.

6)      How prevalent is sarcopenia? Is there a specific reason why this particular condition is targeted?

Response 6: Thanks for your suggestion. Sarcopenia is caused by muscle strength reduction that occurs naturally as humans age. It can cause a decrease in physical activity ability, difficulty in maintaining daily life functions, and an increase in the risk of accidents such as falls. At a time when population aging emerges as a social problem, sarcopenia shows a considerable prevalence and it is considered necessary to pay attention to it. On average, around the globe, it is estimated that 5–13% of elderly people aged 60–70 years are affected by sarcopenia. The numbers increase to 11–50% for those aged 80 or above. According to a paper published in the Journal of the American Medical Directors Association in 2020, the prevalence of sarcopenia in Korea was found to be 21.3% for men and 13.8% for women by applying the 2019 Asian sarcopenia guidelines. Among them, severe sarcopenia was 6.4% in men and 3.2% in women. Since this ratio is huge, our thrust was to target the problem of sarcopenia which might assist the elderly people with a better diagnosis of sarcopenia.

The same is added in the third-last paragraph of the Introduction section in the revised manuscript.  

7)     Why were the training and test sets in GIST not divided by subject number like the other two? Wouldn’t it be better to have the same division method for all three?

Response 7: Thanks for your suggestion. The GIST dataset used leave-one-subject-out cross-validation based on the subject number. Therefore, the dataset does not divide into multiple test subject numbers. Hence, we followed their evaluation method and did not divide test sets with multiple subjects.

8)      Does Table 2 present results of the model before describing the model itself? Wouldn’t it be better placed in the Results section?

Response 8: Yes, Table 2 presents the results of the model. According to your suggestion, we have moved the “Section 3.4; Multiple data inputs along with Table 2” to the “Experimental results section”. In addition, we have added another subsection in the proposed model section, “Ablation study”.  

9)      When discussing gaits, what do the authors mean by being activated? For example, what does “the pelvis and shoulders are activated” mean?

Response 9: Thanks for your suggestion. “The pelvis and shoulders are activated” means this posture gets a high weighting value compared to other joints, which implies that the joints are important for the classification of the gait.

10)   The conclusion is a bit confusing. There are many tables that talk about the accuracy of the model developed in the paper but that is not mentioned in the conclusion. This makes the reader doubt as to what those accuracies were.

Response 10: According to your suggestion, the accuracies for discussed datasets are added in the Conclusion section of the revised manuscript.

11)   Cite a recent report https://doi.org/10.1002/adhm.201901862 on the topic with sentence ´Feature maps were created with features extracted from multiple inputs, and adjacency matrices were defined according to predefined skeletal models ´

Response 11: According to your suggestion, the said reference is added in the revised manuscript.

12)   The authors say that “The main inspiration behind applying the pathological gait classification model is for the diagnosis of sarcopenia” but in the end sarcopenia is never identified. If it is a future work, then the authors are suggested to not mention it in future work and not make it the focus or the main motivation of the paper.

Response 12: Thanks for your suggestion. Inspiration about sarcopenia is added in the 3rd last paragraph of the Introduction section. This inspiration is also added in the Conclusion section of the revised manuscript. In the future, we will try to analyze sarcopenia more accurately and more efficiently, with different aspects and datasets elaborated more precisely in the revised manuscript.    

Round 2

Reviewer 2 Report

Authors addressed the comments raised by reviewers and manuscript is substantially improved. The revised manuscript can be accepted in current form.